**Data Availability Statement:** Data with a very low risk of re-identification and no particular sensitivity ("open access data"), such as aggregated patient

# The Biobanque québécoise de la COVID-19 (BQC19)—A cohort to prospectively study the clinical and biological determinants of COVID-19 clinical trajectories

Karine Tremblay[1,2]ᵒ*, Simon Rousseau[3,4]ᵒ*, Ma'n H. Zawati[5]ᵒ*, Daniel Auld[6], Michaël Chassé[7,8], Daniel Coderre[9], Emilia Liana Falcone[7,10], Nicolas Gauthier[11], Nathalie Grandvaux[12,13], François Gros-Louis[14,15], Carole Jabet[16], Yann Joly[5], Daniel E. Kaufmann[7,13], Catherine Laprise[1,17], Catherine Larochelle[13,18], François Maltais[19], Anne-Marie Mes-Masson[13,20], Alexandre Montpetit[9], Alain Piché[21,22], J. Brent Richards[23,24], Sze Man Tse[25], Alexis F. Turgeon[26,27], Gustavo Turecki[28,29], Donald C. Vinh[30,31], Han Ting Wang[32,33], Vincent Mooser[34], on behalf of BQC19¶

1 Centre Intégré Universitaire de Santé et de Services Sociaux (CIUSSS) du Saguenay–Lac-Saint-Jean, Saguenay, QC, Canada, 2 Department of Pharmacology-Physiology, Medicine and Health Sciences Faculty, Université de Sherbrooke, Sherbrooke, QC, Canada, 3 The Meakins-Christie Laboratories at the Research Institute of the McGill University Heath Centre Research Institute, Montréal, QC, Canada, 4 Department of Medicine, Faculty of Medicine, McGill University, Montréal, QC, Canada, 5 Centre of Genomics and Policy, McGill University, Montréal, QC, Canada, 6 McGill Genome Centre and Department of Human Genetics, McGill University, Montréal, QC, Canada, 7 Department of Medicine, Faculty of Medicine, Université de Montréal, Montreal, QC, Canada, 8 Centre Hospitalier de l'Université de Montréal (CHUM), Montréal, QC, Canada, 9 Génome Québec, Montreal, QC, Canada, 10 Department of Immunity and Viral Infections, Montreal Clinical Research Institute (IRCM), Montreal, Quebec, Canada, 11 CIUSSS du Nord-de-l'Ile-de-Montréal—Hôpital du Sacré-Cœur-de-Montréal, Montreal, QC, Canada, 12 Department of Biochemistry and Molecular Medicine, Faculty of Medicine, Université de Montréal, Montreal, Canada, 13 Centre de Recherche du Centre Hospitalier de l'Université de Montréal (CRCHUM), Montreal, QC, Canada, 14 Centre de Recherche du Centre Hospitalier Universitaire de Québec, Regenerative Medicine Division, Québec, QC, Canada, 15 Department of Surgery, Faculty of Medicine, Université Laval, Quebec, QC, Canada, 16 Fonds de Recherche du Québec Santé, Montreal, QC, Canada, 17 Département des Sciences Fondamentales, Centre Intersectoriel en Santé Durable, Université du Québec à Chicoutimi, Saguenay, QC, Canada, 18 Department of Neurosciences, Université de Montréal, Montreal, QC, Canada, 19 Quebec Heart and Lung Institute, Quebec, QC, Canada, 20 Institut du Cancer de Montréal, Montreal, QC, Canada, 21 Département de Microbiologie et Infectiologie, Université de Sherbrooke, Sherbrooke, Quebec, Canada, 22 Département de Médecine, Service d'Infectiologie, Centre de Recherche Clinique du Centre Hospitalier Universitaire de Sherbrooke, Sherbrooke, Quebec, Canada, 23 Lady Davis Institute, Jewish General Hospital, McGill University, Montréal, QC, Canada, 24 Department of Epidemiology and Department of Human Genetics, Biostatistics and Occupational Health, McGill University, Montréal, QC, Canada, 25 Division of Respiratory Medicine, Department of Pediatrics, CHU Sainte-Justine, Université de Montréal, Montreal, QC, Canada, 26 Centre Hospitalier Universitaire de Québec–Université Laval Research Center, Population Health and Optimal Health Practices Research Unit, Trauma-Emergency-Critical Care Medicine, Québec City, QC, Canada, 27 Division of Critical Care Medicine, Department of Anesthesiology and Critical Care Medicine, Faculty of Medicine, Université Laval, Québec City, QC, Canada, 28 CIUSSS de l'Ouest-de-l'Ile-de-Montréal, Montreal, QC, Canada, 29 Department of Psychiatry, Douglas Mental Health University Institute, McGill University, Montreal, QC, Canada, 30 Division of Infectious Diseases, Department of Medicine, McGill University Health Centre, Montreal, QC, Canada, 31 Division of Medical Microbiology, Department of Laboratory Medicine, McGill University Health Centre, Montreal, QC, Canada, 32 Division of Critical Care Medicine, Department of Medicine, Universite de Montreal, Montreal, QC, Canada, 33 CIUSSS de l'Est-de-l'Ile-de-Montréal, Hôpital Maisonneuve-Rosemont Research Centre, Montreal, QC, Canada, 34 Department of Human Genetics, Faculty of Medicine, McGill University, Montreal, QC, Canada

ᵒ These authors contributed equally to this work.
¶ Membership of the BQC19' contributors is listed in the Acknowledgments.
* karine.tremblay@usherbrooke.ca (KT); simon.rousseau@mcgill.ca (SR); man.zawati@mcgill.ca (MHZ)

data from the different cohorts is available via the BQC19 website (www.BQC19.ca) or can be requested by email at: info@bqc19.ca. Upon completion of the study, the publicly available data will be deposited on a repository with the link provided in the comment section of the article. The stored biological materials will be accessed through a controlled system. Data that has a direct or high risk of re-identification will also go through a tightly controlled access process available at BQC19.ca and described in greater details in the manuscript.

**Funding:** This biobank is financially support by the Fonds de recherche du Québec - Santé (FRQS), Genome Québec and the Public Health Agency of Canada (PHAC). The funders initiated the project to support the research community facing the COVID-19 related sanitary emergency.

**Competing interests:** The authors have declared that no competing interests exist.

## Abstract

SARS-CoV-2 infection causing the novel coronavirus disease 2019 (COVID–19) has been responsible for more than 2.8 million deaths and nearly 125 million infections worldwide as of March 2021. In March 2020, the World Health Organization determined that the COVID–19 outbreak is a global pandemic. The urgency and magnitude of this pandemic demanded immediate action and coordination between local, regional, national, and international actors. In that mission, researchers require access to high-quality biological materials and data from SARS-CoV-2 infected and uninfected patients, covering the spectrum of disease manifestations. The "Biobanque québécoise de la COVID-19" (BQC19) is a pan–provincial initiative undertaken in Québec, Canada to enable the collection, storage and sharing of samples and data related to the COVID-19 crisis. As a disease-oriented biobank based on high-quality biosamples and clinical data of hospitalized and non-hospitalized SARS-CoV-2 PCR positive and negative individuals. The BQC19 follows a legal and ethical management framework approved by local health authorities. The biosamples include plasma, serum, peripheral blood mononuclear cells and DNA and RNA isolated from whole blood. In addition to the clinical variables, BQC19 will provide in-depth analytical data derived from the biosamples including whole genome and transcriptome sequencing, proteome and metabolome analyses, multiplex measurements of key circulating markers as well as anti-SARS-CoV-2 antibody responses. BQC19 will provide the scientific and medical communities access to data and samples to better understand, manage and ultimately limit, the impact of COVID-19. In this paper we present BQC19, describe the process according to which it is governed and organized, and address opportunities for future research collaborations. BQC19 aims to be a part of a global communal effort addressing the challenges of COVID–19.

## Introduction

The coronavirus disease 2019 (COVID-19) is a novel human disease caused by the coronavirus SARS-CoV-2. It was classified as a pandemic by the World Health Organization (WHO) on March 11, 2020. The COVID-19 outbreak is evolving daily, with the total number of deaths now reaching 2,748,737 and confirmed cases surpassing 125,160,255 (WHO, March 26, 2021). Research is essential to better understand the determinants of SARS-CoV-2 infection, the diverse clinical trajectories of infected patients and the determinants of COVID-19 clinical evolution. This work will help clinicians identify individuals at increased risk for complications and poor outcomes in order to adopt appropriate measures to protect them, to help the government take public health measures to control the spread of the infection, and to anticipate and better prepare for future pandemics. Access to high-quality biological materials and data from SARS-CoV-2 infected and uninfected participants is essential for achieving this mission. As part of the solutions to the COVID-19 pandemic, massive investments in coronavirus research have been launched worldwide and biobanks containing biosamples and medical data of individuals having suffered from SARS-CoV-2 infection have become key resources to pursue such research efforts.

In this manuscript, we present the "Biobanque Québécoise de la COVID-19" (BQC19, www.bqc19.ca), a Québec-based biobank infrastructure whose primary objective is to collect and house biosamples and data to support research on COVID-19.

## Infrastructure, study design & methods

### Presentation of the BQC19

On March 26, 2020, the Fonds de recherche du Québec—Santé (FRQS) and Génome Québec announced the launch of a COVID-19 Québec Biobank program, named BQC19. BQC19 is a province-wide initiative to enable the collection, storage and sharing of biosamples and data related to the COVID-19 crisis The Public Health Agency of Canada (PHAC) provided significant additional funds to further support the goals of BQC19.

**Mission.** The mission of the BQC19 is to work in concert with the Quebec network of health institutions of the "Réseau de la santé et des services sociaux du Québec" (RSSS) and academic partners (Research centres and universities) to manage the unique COVID-19 related biological material and data banked at BQC19. The notion of sharing research results is at the heart of the BQC19's mission, and as such, BQC19 has signed the Wellcome Statement on data sharing in public health emergencies, an open-science policy (https://wellcome.org/coronavirus-covid-19/open-data). The BQC19's broad goal is to understand the pathophysiology of COVID-19 and support efforts to discover and develop new biomarkers of disease susceptibility and progression, new or reoriented therapies and vaccines to combat COVID-19. The BQC19 is also directed at enhancing research efforts related to the prevention, treatment, and epidemiological and population management of COVID-19. The BQC19 will stimulate health research and precision medicine initiatives on COVID-19.

**A Quebec hospitals' network.** BQC19 is a multicentric biobanking infrastructure composed of a network of 11 hospitals in Québec and their five partnering academic institutions. All currently participating institutions are presented in **Table 1**. The BCQ19 governance is summarized in **Fig 1** and the composition of each committee is also available on the BQC19 website. BQC19 began its operations on April 1, 2020 and the milestones achieved to date are presented in **Fig 2**. The BQC19 project was approved by the Centre hospitalier universitaire de l'Université de Montréal Institutional ethics review board (**IRB**) [#MP-02-2020-8929, 19.389].

### The BQC19 study design

The BQC19 has been designed as a cohort that includes SARS-CoV-2 PCR negative controls to prospectively study the clinical and biological determinants of COVID-19 clinical trajectories. The BCQ19 conceptual and longitudinal design is illustrated in **Fig 3** and the major components are described in the next subsections.

**Recruitment.** BQC19 includes confirmed COVID-19 (SARS-CoV-2-positive (+)) adults and children who are recruited during a hospital stay (hospitalized cohort). It also includes asymptomatic, mild and moderate ambulatory cases who are recruited one-month post-infection (non-hospitalized cohort). The grading score for severity is based on the WHO Working Group on the Clinical Characterisation and Management of COVID-19 infection [1]. For both groups, SARS-CoV-2 PCR-negative (-) patients are recruited as controls. Thus, in order to be enrolled in the BQC19, the patients must:

1. have undergone a COVID-19 diagnostic test and, for the hospitalized cohort have been admitted to a participating hospital;

2. be willing to participate in optional long-term follow-up;

3. have the capacity to provide informed consent (if the participant is an adult); or have a surrogate decision maker from whom consent can be obtained (in case of incapacity); or have a parental or legal guardian able to provide consent (if the participant is younger than 18 years).

**Table 1. The BQC19 enrolling institutions.**

| Institutions | Investigators |
|---|---|
| Centre hospitalier de l'Université de Montréal (CHUM) | Daniel Kaufmann |
| | Michaël Chassé |
| | Madeleine Durand |
| | Alexandre Prat |
| Quebec Heart and Lung Institute | François Maltais |
| CIUSSS du Saguenay-Lac-Saint-Jean | Catherine Laprise |
| | Karine Tremblay |
| | Luigi Bouchard |
| Centre hospitalier universitaire de Québec (CHUQ)—Université Laval | Alexis Turgeon |
| | Vincent Raymond |
| Centre hospitalier universitaire Sainte-Justine (CHUSJ) | Sze Man Tse |
| | Hugo Soudeyns |
| | Philippe Jouvet |
| | Jean-Sébastien Joyal |
| CIUSSS du Centre-Ouest-de-l'Ile-de-Montréal -Jewish General Hospital | Brent Richards |
| | Jonathan Afilalo |
| McGill University Health Centre (MUHC) | Bruce Mazer |
| | Donald Vinh |
| CIUSSS de l'Ouest-de-l'Ile-de-Montréal—Douglas Mental Health University Institute | Gustavo Turecki |
| | Nadir Hadid |
| | Volodymyr Yerko |
| CIUSSS de l'Estrie—Centre hospitalier universitaire de Sherbrooke | Alain Piché |
| CIUSSS du Nord-de-l'Ile-de-Montréal—Hôpital du Sacré-Cœur-de-Montréal | Nicolas Gauthier |
| | Yiorgos Alexandros Cavayas |
| | Christine Arseneault |
| CIUSSS de l'Est-de-l'Ile-de-Montréal—Hôpital Maisonneuve-Rosemont | Han Ting Wang |
| | Jan Alexis Tremblay |

**Consent considerations.** Informed consent is obtained directly from the adult participant capable of consenting, from a legally authorized representative if the adult is incapable of giving consent or from a parent or legal guardian if aged less than 18 years old. Additionally, assent is obtained from a participating child when appropriate.

Given the high risk of infection for clinical and research staff related to COVID-19, consent is carried out using procedures derived from practices in acute and critical care units and taking into account the particular situation arising from the pandemic.

**Consent procedures.** Each BQC19 enrolling site has established a consent process that reflects the BQC19's standard operating procedures (SOPs, available on www.bqc19.ca). These SOPs address the following specific points: 1) when and where the patient is approached; 2) the procedure to follow when the patient is diagnosed as a SARS-CoV-2 positive or negative PCR result; 3) the timing and nature of sampling (including data) depending on whether the patient is diagnosed as SARS-CoV-2 positive or negative and whether the patient is hospitalized, and; 4) the time period over which recruitment is to be conducted. The SOPs developed for BQC19 provide details on each of these points targeted to each facility. To ensure consistency across BQC19 and to ensure that procedures are harmonized, consent processes established by the BQC19 participating establishments must follow two fundamental principles: 1) respect of the autonomy of the participants (taking into account their state of health) according

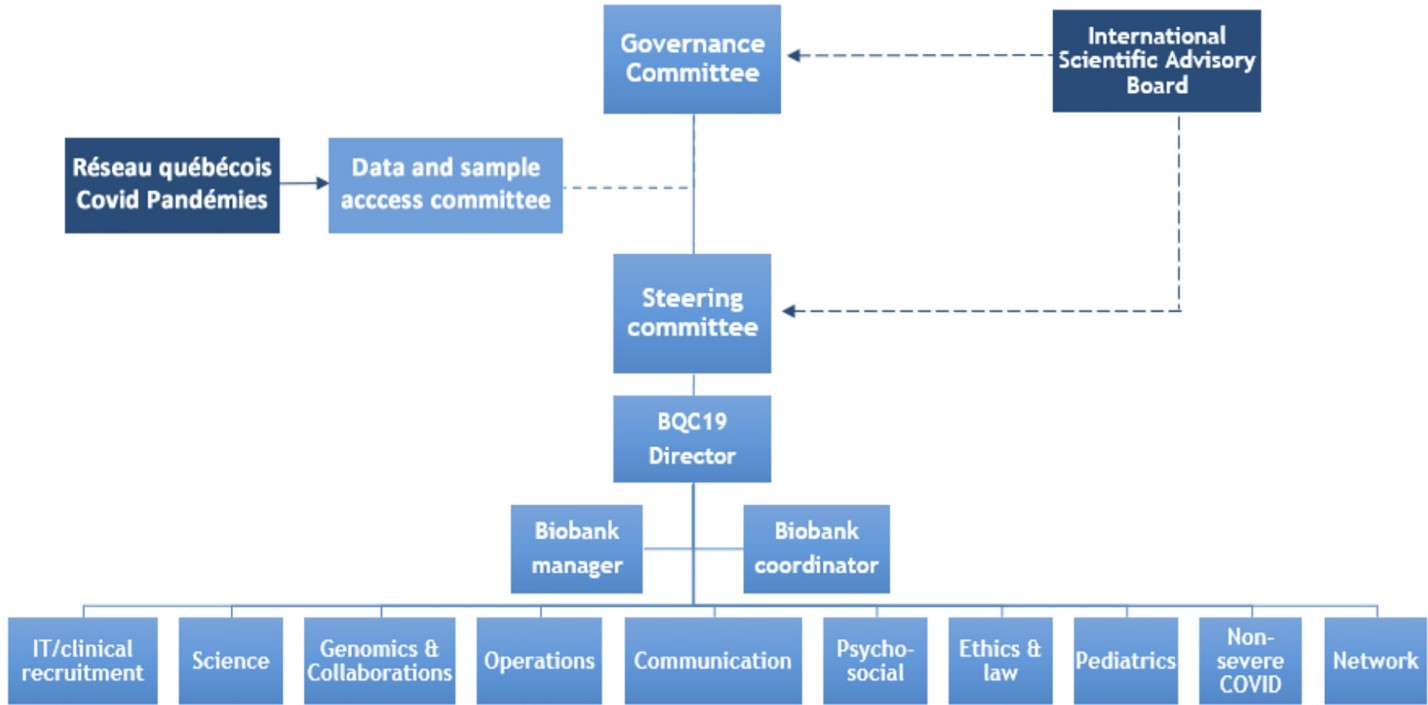

**Fig 1. BQC19 organizational chart.** BQC19 is a biobank with its own management and governance structure. The governance includes a Governing Committee, a Steering Committee, an independent Data and Sample Access Committee, and an international Scientific Advisory Board (**Antoine Flahault**, MD, Ph.D., Director, *Institut de santé globale*, *Université de Genève*, Switzerland (President); **Andrew D. Badley**, MD, Principal Investigator, Mayo COVID19 Biobank, Rochester, Minnesota, USA; **Mark Daly**, Ph.D., Co-director, Program in Medical and Population Genetics, Broad Institute, Cambridge, Massachusetts, USA; **Daniel Douek**, MD, Ph.D., Chief of the Human Immunology Section, NIAID, NIH, Bethesda, Maryland, USA; **Mette Hartlev**, LLM, Ph.D., LLD, Professor, *Centre for Legal Studies in Welfare and Market*, Denmark; **Gary Kobinger**, Ph.D., Canada Research Chair in immunotherapy and innovative vaccine platforms, *Centre de recherche du CHU de Québec*, *Université Laval*, Quebec, Canada; **Rosanna Peeling**, Ph.D., London School of Hygiene and Tropical Medicine, London, UK, Professor/Chair of Diagnostics Research, Director of the International Diagnostic Centre (IDC)). It also includes several sub-committees responsible for mandates ranging from scientific priorities to communication and ethical, legal and social issues. The governance of BQC19 is framed in its Management Framework. Terms of References for accessing samples and data collected within the framework of BQC19 are being completed.

to provincial legal and research ethics standards and 2) ensure the safety of all stakeholders involved at all times. Additional information can be found in (Appendix 1- Consent in S1 File).

## The BQC19 sample collection

**BQC19 collected samples and availability.** For adults who have consented to participate in BQC19 and are hospitalized, 48 ml peripheral venous blood samples are drawn at up to five different timepoints during the participant's clinically indicated blood work. Blood samples are collected when possible: on the day of recruitment (T0); on Day 2 (Q2); on Day 7 (Q7); on Day 14 (T14); and on Day 30 (T30) or at the first available time if the window was missed. For participants who were discharged from hospital, an additional 60 mL of blood is drawn at each of the follow-up visits scheduled approximately at months 1, 3, 6, 12, 18 and 24 following hospital discharge (outpatient or home). For those participating to follow-up, blood is not necessarily collected as part of standard care and a maximum of 200 ml of blood per month can be collected. For adults who have consented to participate in BQC19 but have not been hospitalized, a 60 mL of blood is drawn at each of the scheduled follow-up visits approximately in months 1, 3, 6, 12, 18 and 24. For these participants, blood is also not necessarily collected as part of standard care and a maximum of 200 ml of blood per month can be collected. For both

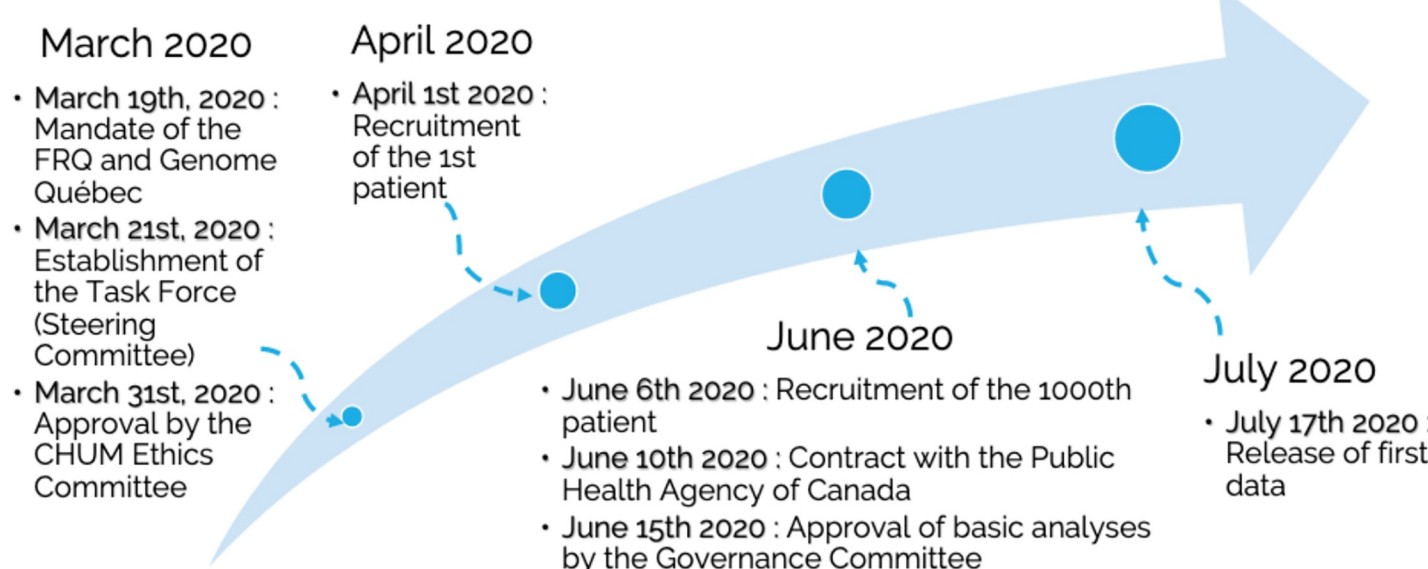

**Fig 2. BQC19 milestones.** The key BQC19 milestones achieved since the start of its mandate received on March 19[th], 2020 leading to the release of the first set of data (July 17, The key BQC19 milestones achieved since the start of its mandate received on March 19[th], 2020 leading to the release of the first set of data (July 17, 2020).

cohorts, follow-up visits are optional and participants may opt to agree only provide clinical information if they do not wish to donate blood samples. For children, the adult protocol is followed but the total volume is determined according to the weight of the child. If the parent

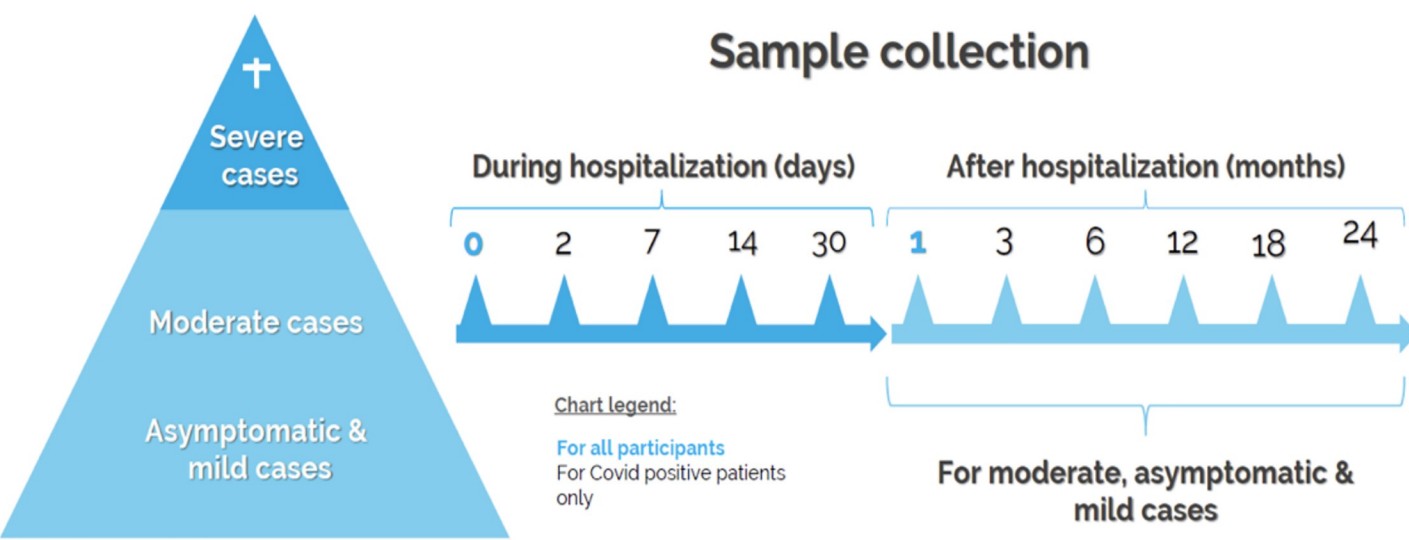

**Fig 3. BQC19 study design.** Schematic representation of the BQC19 study design. For hospitalized patients (hospitalized cohort) samples are collected during hospitalization at the days indicated (darker blue) and following hospitalization at the months indicated (paler blue). For asymptomatic, mild and moderate disease out-patients (non-hospitalized cohort), samples are collected at the indicated time points (pale blue). Samples to be collected by all participants (pale blue) and COVID-19 + only (black).

**Table 2. BQC19 available biosamples.**

| Type of samples | Total [a] |
|---|---|
| DNA isolates [b] | 2,726 |
| RNA isolates [b] | 3,794 |
| Plasma | 3,930 |
| Serum | 1,244 |
| PBMCs | 3,198 |
| Total | 14,892 |

PBMCs = Peripheral blood mononuclear cells.

[a]As of March 19, 2021.

[b] Include all expected samples collected from whole blood (DNA) or PAXgene® (RNA) tubes.

refuses a blood draw for research, their permission is obtained to recuperate leftover samples from the clinical laboratory. Depending on the possibilities for blood samples to be processed, the BQC19 sample collection includes: 1 PAXgene® RNA tube, 4 Acid Citrate Dextrose (ACD) tubes and 1 red-capped tube (serum) from each participant at each visit where blood sample are drawn. These samples allow DNA/RNA, plasma, serum and peripheral blood mononuclear cells (PBMCs) isolation. All tubes are kept at room temperature before processing; samples are processed rapidly after phlebotomy, ideally < 6 hours; <12 hours is fine for most assays; >12h: a number of functional assays will become less reliable. The retrieval time between venipucture and sample handling is documented. The complete sample processing is available in the **Appendix 2 in S1 File**. The BQC19 stored biosamples inventory is presented in **Table 2**.

## Biosamples pre-analytical quality control

In order to ensure consistency in the preparation of biosamples collected for BQC19, all participating sites use the same SOPs (available at BQC19.ca), clearly detailing the exact protocol to be followed, including the type of primary container to be used (**Table 3**). In terms of pre-analytical quality assessment, we document the following information for each sample collected: date of collection, location of sample, associated barcode(s), SOPs used, time elapsed to biobank, precisions/explanations on delays, name of sample's handler. Our protocol details that samples should ideally be processed in less than 6h from collection, the actual time (pre-centrifugation time) is recorded for each sample in the biobank management software. In addition, for PBMCs we document cell concentration, number of cells in sample (total cells count),

**Table 3. BQC19 biosample pre-analytical quality information.**

| Type of sample | Type of primary container | Pre-centrifugation delays [a] | Centrifugation | Long-term storage |
|---|---|---|---|---|
| Whole blood | ACD, SED, SHP | <1h | | -80˚C (2ml tubes, barcoded) |
| RNA | PAX | <2h | | overnight RT, 24h at -20˚C, -80˚C long term |
| Plasma | ACD | <3h between 1-6h | 850 X g, 10 min RT, no Brake | -80˚C (1.5ml cryotube) |
| Serum | CAT ou SST | >6h | 2000 X g, 10 min RT, Brake | -80˚C (1.5ml cryotube) |
| PBMCs | ACD | unknow and specified delay | 300 X g, 4˚C, no Brake | Liquid nitrogen (sterile cryotubes) |

ACD = Acid citrate dextrose tube; CAT = Serum tube without clot activator; PAX = PAXgene® tube; PBMCs = Peripheral blood mononuclear cells; RT = Room temperature; SED = Sodium EDTA tube; SHP = Sodium heparin tube; SST = Serum separating tube with clot activator.

[a] Indicated values are captured for each biosamples in biobank management software.

freezing medium; for nucleic acids isolations, we document extraction date, extracted from [biological specimen], extraction method, diluent, absorbance measurements at 260nm, 280nm and 320nm, 260/280 ratio, dilution factor, concentration, total amount.

## Access to BQC19 samples and data

Usage of BQC19 biosamples and data is only possible if aligned with a participant's consent. This is made possible by ensuring that the access process as well as the terms and conditions of any future use of data and samples respect the general permissions consented to by participants. Access must respect the rights, interests and expectations of the BQC19 participants and must support the research to which they initially consented, consistent with the mission of the BQC19. Access to data, a renewable resource, is planned in a manner that allows rapid data use by applicants to meet urgent research needs associated with COVID-19. An expedited assessment process is in place for requests to access data alone. Access to biological samples, a limited resource, requires additional steps.

**Open and controlled access.** Data with a very low risk of re-identification and no particular sensitivity ("open access data"), such as aggregated patient data from the different cohorts will be made publicly available on the BQC19 website. For the stored biological materials, they will be accessed through a controlled system. Data that has a direct or high risk of re-identification will also go through a tightly controlled access process. Access to the BQC19 resources complies with the processing principles described below.

**Principles guiding access to BQC19 data and biosamples.** Requests from investigators who wish to access BQC19 samples and data are reviewed by the independent Biobank Access Committee. The eligibility criteria to apply for access are summarized in **Fig 4** and the procedure in **Fig 5**. The details can be found in (**Appendix 3- Access in S1 File**). A registry of all projects that have benefited from biomaterial and data of BQC19 is maintained and will be made available to the research community and the general public on the BQC19 website.

# Results & discussion

## Participant's profiles (April to March 2021)

**Enrollment statistics.** For the hospitalized cohort, we report a 75.4% acceptance rate (2,1271 out of 2,878 invited to participate excluding patients who were discharged or scheduled to be discharged, deceased, incapacited, or admitted to care units without planned blood sampling; total form eight sites). In the non-hospitalized cohort, we report an acceptance rate of 80.2% (616 out of 768 invited to participate; total from five sites). The higher success rate in the non-hospitalized cohort may be explained by different enrolling strategies across sites (e.g. two of the sites used public advertising which includes a voluntarism bias). In term of dropout rates, we report 3.4% in the hospitalized cohort (73 dropouts out of 2171 enrolled participants) and a 0.5% rate in the non-hospitalized cohort (3 dropouts out of 616 enrolled participants). Finally, for both cohorts, reported reasons for refusal or study dropout include: "no interest", "no benefit", "don't believe in research purposes", "no time for follow-up", "surrogate refusal", "health related reasons/age", "SARS-CoV-2 negative patient who though their participation wasn't important", "fear about the future uses of their data (or their children's data)", "parents' fear of harming their children", "unwillingness to move or to give more blood for follow-up visits", "communication/understanding issues", "difficulty in taking blood samples", and "overburdened by hospitalization and their clinical follow-up/worried enough".

**BQC19 participants' characteristics and available data.** As of March 19, 2021, 2,787 participants have consented to participate to the BQC19. However, quality controlled data is currently available for a total of 2,300 enrolled participants (2,256 adults and 44 children recruited

# Eligibility criteria for requesting access

- **Eligibility criteria for submitting an access request**

  - Academic Researcher in Canada

  - Academic Researcher outside of Canada

  - Researcher in the private sector

- **Access will be allocated in a fair, non-discriminatory, objective and transparent manner to all researchers, regardless of their research discipline**

# Evaluation criteria

- **Criteria for access to data**

  - Request for access must be inline with BQC19's mission available in the management framework and on BQC19's website.

  - Access must respect the rights, interests and expectations of BQC19's participants.

  - The researchers will have explicitly stated that they will not attempt to re-identify participants and protect the confidentiality of the data transferred.

  - If required, measures describing how the risk of re-identification will be minimised.

- **Additional criteria for accessing biological materials:**

  - Originality of the research question in relation to projects already in progress or projects that are subject of peer-reviewed publications

  - Value of data returned to BQC19

  - Potential risk of biosample's depletion

  - Demonstrated scientific validity, consistent with BQC19 mission: The use of the Biobank resources must help to maximize scientific, clinical and social advantages.

  - Feasibility of the project (validation techniques in the applicants' laboratories. Adequate financial support to achieve the objectives). Samples should not be used as development material apart from exceptional cases directly related to BQC19's mission

  - Expertise of the team in the specific field

**Fig 4. Eligibility and evaluation criteria for BQC19 access.** The figure lists the general eligibility and evaluation criteria to obtain access to BQC19 biological material and data.

between April 2020 and March 2021). A total of 1,635 confirmed SARS-CoV-2 PCR positive cases (789 males and 846 females) aged between 0 and 104 years (adults mean of age of 59.2± standard deviation of 19.6 years; children mean age of 7.3±7.0 years) and 644 SARS-CoV-2

PCR negative controls (335 males and 330 females) aged between 0 and 102 years (adults mean age of 62.5±20.1 years; children mean age of 6.4±6.7 years) were included. Among all subjects, 1,716 (1,110 SARS-CoV-2 PCR positive cases and 596 negative controls) are part of hospitalized cohort while 584 (515 SARS-CoV-2 PCR positive cases and 69 negative controls) are part of non-hospitalized cohort; their distribution according to follow-up visits is presented for both cohorts in **Fig 6**. For the hospitalized cohort, where participants have been enrolled at the time of hospital admission (Day 0), the full sample sets currently available at each time-point are shown in **Fig 6A**. For the non-hospitalized cohort, the full sample sets are available for all participants at Day 180 post-infection (**Fig 6B**). However, in this cohort, some participants may have been enrolled at Day 30 or Day 90 post-infection.

The relevant demographic, clinical and pharmacological variables for each participant are collected following a chart review documented in a case report form (CRF) (available at www. bqc19.ca). The participants currently included in BQC19's database are distributed in four Québec Health Regions: 1808 (78.6%) from Montréal (five enrolling sites); 291 (12.7%) from Estrie (one enrolling site); 144 (6.3%) from the Saguenay-Lac-Saint-Jean (one enrolling site); and 57 (2.5%) from the Capitale-Nationale (two enrolling sites). This is not an accurate reflection of the demographic representation of the province's population, which was not the goal of this biobank, but rather to recruit participants as quickly as possible, to support research during the sanitary emergency.

The consent to BQC19 participation allows access to participants' medical chart as well as information contained in the Quebec public health administrative databases (e.g. the "Institut de la Statistique du Québec (ISQ)" or the "Laboratoire de santé publique du Québec").

## BQC19 key features

In this section, we outline a few key features of BQC19 that may be useful to the research community in taking advantage of its resources.

**An evolutive biobank management framework.** The management framework is at the core of any biobank initiative. It defines key structural and procedural elements associated with resources. These complex documents need significant forethought and usually require a considerable time investment, an element that was not available to the BQC19 since the goal was to begin recruitment at the dawn of the first wave of COVID-19 hospitalizations in the spring of 2020. Given the urgency of the situation, this management framework was developed and approved in several distinct phases to both address the urgent need to start operation, while respecting the core values of ethics and transparency. The first phase focused on enabling recruitment, followed by governance and access. This process allowed BQC19 to be receptive to a shifting on-the-ground reality, both scientifically and ethically, and to enable the management framework to rapidly adapt to reflect these realities while at the same time remaining innovative, anticipatory and forward-looking. This iterative procedure was only possible through a tight and dynamic collaboration with IRBs of the hospitals participating in the BQC19. For more details, the BQC19 management framework is available on BQC19 website.

**Standardization across sites.** A key requirement of a multicentric project is the uniformization of processes across all recruiting sites using the same SOPs. This allows studies to be performed on a greater number of samples and to compare disease profiles across regions. This is particularly important to limit pre-analytical issues for "omics" analyses.

**PBMCs collected longitudinally.** Isolating PBMCs from blood is a resource intensive procedure. However, there is great value added by having access to frozen PBMCs to study the activity of the immune system during COVID-19. We have favored the collection of PBMCs in hospitalized and ambulatory patients, including longitudinal sampling at multiple days

A.  Process for access to data only

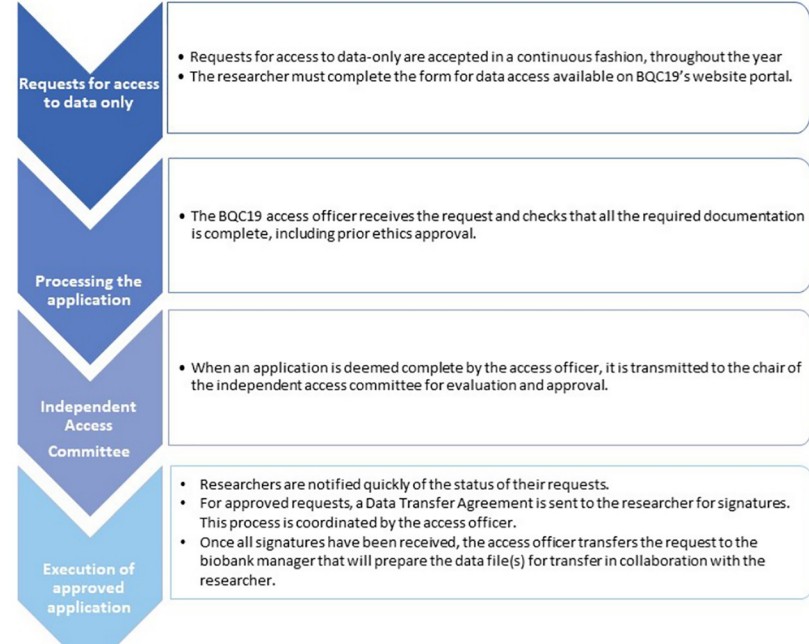

B.  Process for access to biological material and data

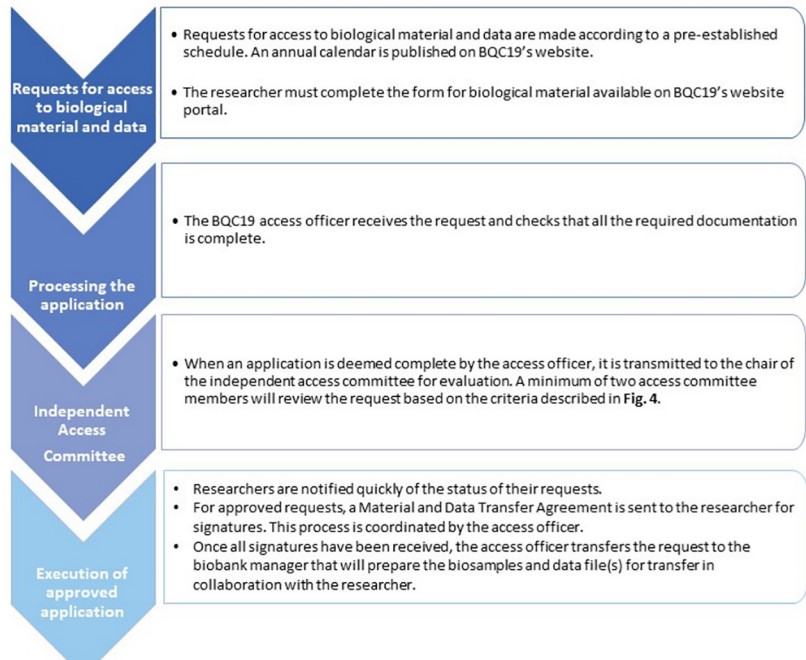

**Fig 5. Flow chart of BQC19 access process.** The chart illustrates the steps required to gain access to BQC19 data only (A) or biological material and data (B).

## A. Hospitalized cohort

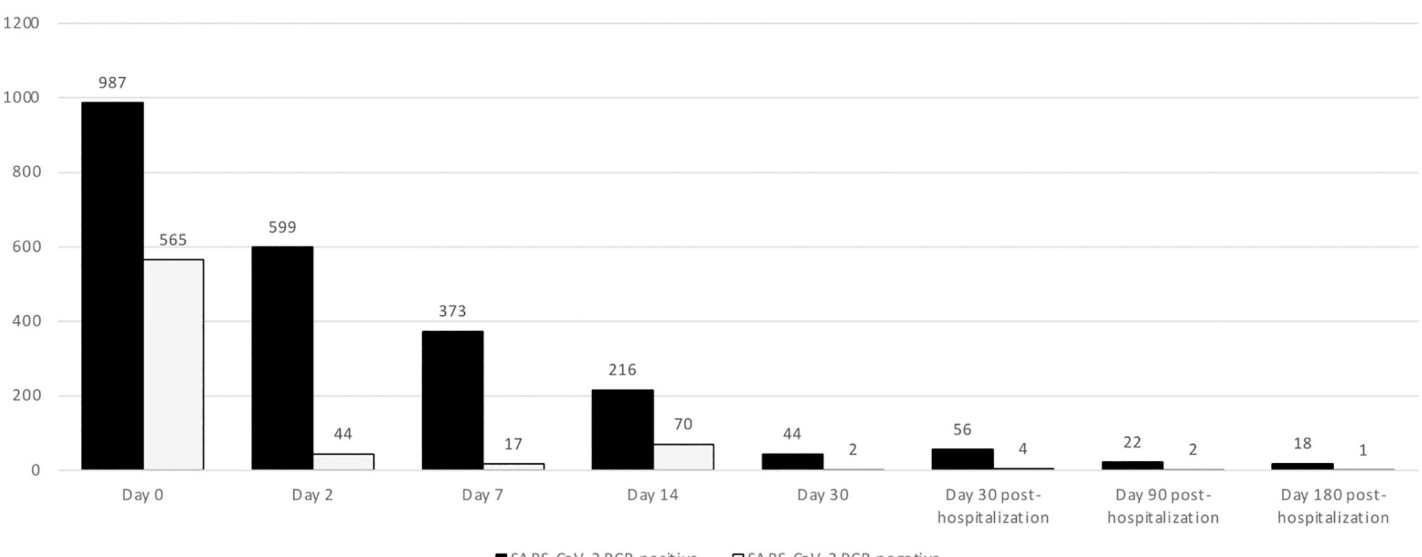

## B. Non-hospitalized cohort

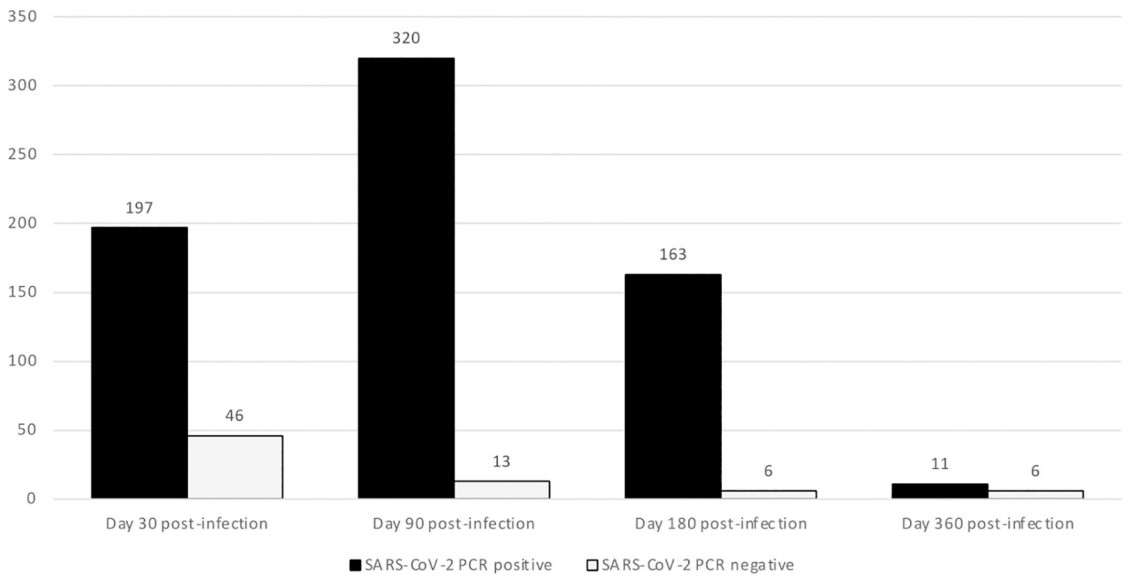

**Fig 6. Longitudinal distribution of BQC19 participants.** The number of participants to the hospitalized cohort (A) and to the non-hospitalized cohort (B) at each time point of sampling is given above each bar. Of note, for the hospitalized cohort, follow-up visits are calculated after patient hospital discharge and for the non-hospitalized cohort, follow-up visits are calculated after patients diagnosis (PCR confirmed). Black bars represent SARS-CoV-2 PCR positive cases while grey bars represent SARS-CoV-2 PCR negative controls. Data as of March 19, 2021.

following recruitment. While not all sites were able to do this collection, we nevertheless have multiple longitudinal cryopreserved PBMCs samples, a distinctive valuable resource that will help to better understand the role of circulating immune cells in SARS-CoV-2 infection and the development of COVID-19. In addition, PBMCs could also be reprogrammed into induced pluripotent stem cells (IPSC) to generate in vitro models, such as, organoids to better understand the disease pathophysiology.

**Population profile.**   The majority of data and samples collected to date are from the Montreal region, which was one of the worst hit cities in Canada during the spring 2020 wave. Montreal is a cosmopolitan city, with a multicultural and multiethnic population. The BQC19 multicentric design allows, in addition, the collection of data and samples from participants of other regions of the Québec province. This wide coverage of Quebec's population may take advantage of the inclusion of some population profiles that are much less diverse from a genetic point of view. For example, the BQC19 includes participants from the Saguenay-Lac-St-Jean region, well-recognized as a founder population [2–4] that has been demonstrably useful in genetic studies of Mendelian traits [5]. Moreover, while the current framework is targeted to the adult population, BQC19 is integrated with a multicentric pediatric biobank led by one of Montréal's pediatric hospitals, the Centre Hospitalier Universitaire Sainte-Justine (CHUSJ). This means that within BQC19, access to pediatric data and biosamples is also available, and extension of the recruitment of maternal biosamples and data is planned.

**Core analyses.**   In view of the limited availability of biosamples and to support the greatest accessibility to analytical/experimental data to the research community, the BQC19, with the support of its funding agencies, has established a plan for core analyses of a large subset of biosamples that include performing whole genome sequencing, genome-wide association studies, transcriptomics, proteomics, metabolomics, circulating inflammatory marker profiling and serology (titers of SARS-CoV-2 antibodies, including neutralization activities) using the same technologies for all samples. These analyses are summarized in **Table 4**. They will be directly integrated into BQC19 database and will be available to all authorized researchers to access.

**Open science.**   As stated, the core of the BQC19 mission is the sharing of data with the entire research community in respect of its ethical and legal obligations. This includes the requirement that all users return analytical and experimental data obtained with BQC19 biosamples to the biobank for other researchers to access. This is a condition of BQC19 usage and is an investment in its future wealth as a sustainable resource. The BQC19 fully subscribes to the Statement of data sharing in public health emergencies (https://wellcome.org/coronavirus-covid-19/open-data).

## Future directions

**Recruitment.**   Following the first wave of the pandemic, additional financial support from the Public Health Agency of Canada was secured to support the expansion of its activities to non-hospitalized participants. This phase of recruitment has begun and aims to add asymptomatic or mild to moderate cases of COVID-19 to the BQC19 resources. Moreover, as of writing of this manuscript, infections are on the rise again in Quebec, and BQC19 pursues its recruitment for both out- and in-patients. The second wave is characterized by a much higher proportion of confirmed cases in individuals in the 20–49 age group (https://www.quebec.ca/en/health/health-issues/a-z/2019-coronavirus/situation-coronavirus-in-quebec/#c63039). Recruitment of this population will broaden the age spectrum within BQC19 and enable more comprehensive studies looking at COVID-19 throughout the life span. This is in addition to the current integration with the pediatric arm of BQC19.

**Networking.**   Finally, a key to overcoming challenges posed by the current pandemic is open collaboration. In addition to its policy on open science and making all biobank documentation freely available via its website, BQC19 is actively pursuing partnership with other initiatives at national and international levels. This includes networking with other biobanking initiatives in Canada (Alberta, Ontario, New Brunswick and Nova Scotia) like CanCov (https://cancov.net) as well as with large population cohorts, such as CARTaGENE (www.cartagene.qc.ca) and the Canadian Longitudinal Study on Aging (CLSA, www.clsa-elcv.ca).

**Table 4. BQC19 planned core analyses.**

| Type of analysis | Objective of the analysis |
|---|---|
| **Genome-wide genotyping & Whole genome sequencing** | Identification of genetic variants in the host genome and genetic variations such as changes in the copy number of certain genes (genome-wide sequencing) as well as common genetic variations across the genome (genome-wide genotyping) associated with COVID-19. The results will allow studies on the susceptibility and risk of developing a severe form of the disease. |
| **Viral genome sequencing** | This analysis will provide a better understanding of the propagation of the pandemic n and the different strains of virus identified. These data can also be correlated with disease severity and immune responses as well as with host genome sequencing. |
| **Proteomic (1)** | The simultaneous measurement of approximately 5000 proteins using the SomaScan technology from SomaLogic in the collected samples shall provide data to predict the risk of disease progression. This technology was chosen because of the large number of proteins measured in a single sample. |
| **Proteomic (2) Circulating markers** | This approach is complementary to SomaScan above and will allow the measurement of established markers of inflammation/disease activity using a very specific and sensitive technique. These data will allow a better understanding of the biology of patient responses to disease and help guide future treatment. |
| **Core hospital laboratory analysis for outpatients (non-hospitalized cohort)** | These analyses will allow basic blood tests to be performed on non-hospitalized patients and will provide important data for research on participants in both cohorts. This includes baseline values for liver, heart and kidney damage, as well as standard inflammation parameters. |
| **Metabolomic** | Establishing the plasma metabolome will complement the proteomic data and will enhance capacity to identify/predict individuals at risk of developing severe disease and favouring a deeper understanding of the molecular pathways regulating the various clinical trajectories. |
| **Serology** | This analysis will allow for very detailed and quantitative measurement of specific antibodies against the SARS-CoV-2 virus in affected patients, well beyond standard serological tests, as well as the ability of these antibodies to neutralize the virus. This will help guide research on the immune response of patients to COVID-19, a key element in the management of the disease. |
| **Transcriptomic** | Transcriptomic gene signatures have been associated with other viral diseases with cellular and immune responses, the pathogenesis of the disease and the trajectory of infection. Transcriptomic analyses performed on participants' RNA extracted from whole blood will generate important data in this area for COVID-19. |

These networking efforts are key in enhancing the scientific community's research capacity. Moreover, via collaboration with nation-wide COVID-19 genomic initiatives in Canada, such as HostSeq (www.cgen.ca/project-overview) or VirusSeq (www.genomecanada.ca/en/cancogen/cancogen-virusseq), BQC19 aims to provide for as many participants as possible, the host and SARS-CoV-2 genomic data isolated by the "Laboratoire de Santé Publique du Québec" since the beginning of COVID-19 testing in Québec. This integration will create a comprehensive and rich data bank, enabling innovative studies on host-pathogen interactions at the genetic level.

## Conclusion

BQC19 is a COVID-19 dedicated biobank which has been designed to prospectively capture data and samples from a large number of SARS-CoV-2 PCR positive and negative controls

during the COVID-19 pandemic. We have already approved access to data or biological material to more than a dozen investigators in the first few months of operations. By providing access to the research community to clinical data as well as data derived from in-depth multi-omic analyses on the first 2000 samples, we are forecasting (and encouraging) an exponential increase in requests of this valuable and non-depletable resource. BQC19 is a critical infrastructure to study the molecular and clinical determinants of COVID-19 susceptibility, severity and outcomes.

## Supporting information

**S1 File.**
(DOCX)

**S2 File.**
(DOCX)

## Acknowledgments

Authors are grateful to all participants for their essential and valuable contribution. We would also like to acknowledge the financial support from the Fonds de recherche du Québec—Santé (FRQS), Genome Québec and the Public Health Agency of Canada (PHAC). In addition, we would like to thank Rémi Quirion (Québec chief scientist, FRQ), Serge Marchand (Génome Québec) and Pascal Michel (scientific adviser, PHAC). Special thanks to BQC19 staff, Pascale Léon (manager), Mylène Bertrand (coordinator) and Doris Ransy (access officer). Michael Lang, Academic Associate, Centre of Genomics and Policy, McGill University, for his editorial contribution.

The BQC19 group is led by Dr. Vincent Mooser (BQC19 director, vincent.mooser@mcgill.ca) is composed of numerous health professionals, researchers and other highly qualified personnel contributors listed below according to their primary affiliation. Each partner or institution are listed in alphabetical order as well as each contributor (by last name). *Centre hospitalier universitaire de Québec*: François Belleau, David Bellemare, Olivier Costerousse, Philippe Després, Ève Dubé, Martin Godbout, Samantha Jacques, Patrick Laplante, Vincent Raymond, Serge Rivest, Hugo Noël-Thiboutot. *Centre hospitalier universitaire de l'Université de Montréal*: Pascale Arlotto, Fatna Benettaib, Dounia Boumahni, Nathalie Brassard, Marie-Ève Cantin, Annie Chamberland, Madeleine Durand, Camille Craig, Andrés Finzi, Ali Ghamraoui, Nakome N Guissan, Juliana Lanza, Stéphanie Matte, Marc Messier-Peet, Livia Pinheiro-Carvalho, Alexandre Prat, Vincent Poitout, Maya Salame, Martine Sauvé. *Centre hospitalier universitaire de l'Université de Sherbrooke*: William Fraser, Annie Laventure, Christine Rioux-Perreault, Karine Tremblay. *Centre hospitalier universitaire Sainte-Justine*: Isabelle Boucoiran, Lucy Clayton, Sylvie Cossette, Mariana Dumitrascu, Mary-Ellen French, Simon Jacques-Ricard, Philippe Jouvet, Jean-Sébastien Joyal, Vincent Laguë, Ariane Larouche, Jacques Michaud, Hugo Soudeyns. *Centre intégré universitaire de santé et services sociaux du Saguenay-Lac-Satin-Jean*: Christian Allard, Donald Aubin, Audrey Baril, Jean-François Betala-Belinga, Jean-Sébastien Bilodeau, Cynthia Bouchard, Luigi Bouchard, Isabelle Boulianne, Marie-Ève Dubeau, Marco Duchesne, Martin Fortin, Hélène Gagné, Ann-Lorie Gagnon, Christine Gagnon, François Gagnon, Maude Gagnon, Caroline Giroux, Doria Grimard, Sharon Hatcher, Guillaume Jourdan, Julie Labbé, Marlène Landry, Julie Larouche, Vanessa Larouche, Myriam Lavoie, Julie Létourneau, Kara Létourneau, Nadia Mior, Louise Poirier, Stéphanie Potvin, Marie-Andrée Régis, Roger Savard, Ruth St-Gelais, Mélanie Tanguay, Nancy Tremblay, Véronick Tremblay, Karine Truchon. *Hôpital Maisonneuve-Rosemont/*

*Centre intégré universitaire de santé et services sociaux de l'Est-de-l'Ile-de-Montréal*: Denis-Claude Roy, Martin Sirois, Danae Tassy, Jan Alexis Tremblay. *Hôpital Sacré-Cœur/Centre intégré universitaire de santé et services sociaux du Nord-de-l'Ile-de-Montréal*: Christine Arsenault, Kim Beauchesne, Sylvie Beaulieu, Paul Bergeron, Mariane Bertagnolli, Caroline Bouchard, Yiorgos Alexandros Cavayas, Marie-Laure Dablaka, Anatolie Ducas, Mathilde Duplaix, Marc-André Gagné, Kim Gilbert, Julie Hammamji, Anne-Marie Ledoux, Claudia Ménard, Sébastien Saucier, Daniel Sinnett, Carla Sterlin, Virginie Williams. *Institut universitaire en santé mentale Douglas/Centre intégré universitaire de santé et services sociaux du Ouest-de-l'Ile-de-Montréal*: Sylvanne Daniels, Nadir Hadid, Amine Saadi, Volodymyr Yerko. *Jewish General Hospital*: Tala Abdullah, Olumide Adeleye, Darin Adra, Jonathan Afilalo, Marc Afilalo, Zaman Afrasiabi, Noor Almamlouk, Amanda Babitt, Gerry Batist, Stéphane Benhamou, Bessy Bitzas, Kathleen Blagrave, Levon Boodaghians, Mariem Bouab, Bluma Brenner, Janet Chan, Jesse Chevrier, Justin Cross, Bianca D'Iorio, Gaby Dipancrazio, Vince Forgetta, Melyssa Fortin, Diane Gaudreau, Biswarup Ghosh, Celia Greenwood, Charlotte Guzman, Amanda Hakala, Gay Hazan, Danielle Henry, Esther Kang, Laetitia Laurent, Geneviève Lefebvre, Melanie Leung, Chen Liang, Rod McInnes, David Morrison, Alexander Ni, Kimchi Nofar, Marianna Olegovna Orlova, Gabriel Ouellette, Damon Palmer, Louis Petitjean, Nardin Rezk, Jennifer Robinson, Lawrence Rosenberg, Myriam Sahi, Erwin Schurr, Lingqiao Song, Samy Suissa, Phil Troy, Christine Tselios, Branka Vulesevic, Xiaoqing Xue, You Jia Zhong. *Laboratoire de Télématique Biomédicale*: Mina Dligui, Éric Rousseau, Yvan Fortier. *McGill University*: David Anderson, Alexandre Belisle, Ariane Boisclair, Guillaume Bourque, David Buckeridge, David Bujold, Elizabeth Caron, Martha Crago, Corinne Darmond, Ksenia Egorova, Tim Evans, Philippe Gros, Peter Ho, Tony Kwan, David Langlais, Mark Lathrop, Claire Le Moigne, Pierre Lepage, Markus Munter, Guillaume Lesage, Kristina Öhrvall, Antoine Paccard, Ioannis Ragoussis, Maryam Rajaee, Janick Saint-Cyr, Rob Sladek, Alfredo Staffa, Patrick Willett. *McGill University Health Centre*: Maria Bazan, Nick Bertos, Julie Bérubé, Miguel Burnier, Melissa Gaudet, Marie Hirtle, Marianne Issac, Bruce Mazer, Geoffrey McKay, Andrea Mogas, Naiana Muntini, Brigitte Paquet, Hansi Peiris, Anna Perez, Ciriaco Piccirillo, Rhyan Pineda, Lucie Roussel, Sandeep Vanamala, Rosemary Wagner. *National Microbiology Lab*: Guillaume Poliquin. *Quebec Heart and Lung Institute*: Sabrina Biardel, Jamila Chakir, Stéphanie Gormley, Philippe Joubert, Christine Racine, Denis Richard. *Réseau Québécois COVID-19 –Pandémie*: Vincent Dumez, Amélie Forget, François Lamontagne. *Touché Créations*: François Brouillet. *Université du Québec à Chicoutimi*: Stéphane Allaire, Jessica Bélanger, Anne-Marie Boucher-Lafleur, Marie-Ève Bradette-Hébert, Yves Chiricota, Frédéric Desgagné, Claire Fournier, Sandra Lessard, Marie-Josée Roy, Claude Thibeault.

## Author Contributions

**Conceptualization:** Karine Tremblay, Simon Rousseau, Ma'n H. Zawati, Daniel Auld, Michaël Chassé, Emilia Liana Falcone, François Gros-Louis, Carole Jabet, Yann Joly, Daniel E. Kaufmann, Catherine Laprise, Catherine Larochelle, Anne-Marie Mes-Masson, Alexandre Montpetit, Alain Piché, J. Brent Richards, Sze Man Tse, Donald C. Vinh, Vincent Mooser.

**Data curation:** Karine Tremblay, Simon Rousseau, Daniel Auld, Michaël Chassé, Emilia Liana Falcone, Daniel E. Kaufmann, Alain Piché, J. Brent Richards, Sze Man Tse, Donald C. Vinh, Vincent Mooser.

**Formal analysis:** Karine Tremblay, Simon Rousseau, Daniel Auld, Michaël Chassé, Emilia Liana Falcone, Daniel E. Kaufmann, Alain Piché, J. Brent Richards, Sze Man Tse, Vincent Mooser.

**Funding acquisition:** Karine Tremblay, Simon Rousseau, Ma'n H. Zawati, Daniel Auld, Michaël Chassé, Daniel Coderre, Carole Jabet, Daniel E. Kaufmann, Catherine Laprise, Alain Piché, J. Brent Richards, Sze Man Tse, Donald C. Vinh, Vincent Mooser.

**Investigation:** Karine Tremblay, Simon Rousseau, Ma'n H. Zawati, Daniel Auld, Michaël Chassé, Emilia Liana Falcone, Nicolas Gauthier, François Gros-Louis, Daniel E. Kaufmann, Catherine Laprise, François Maltais, Alain Piché, J. Brent Richards, Sze Man Tse, Alexis F. Turgeon, Gustavo Turecki, Donald C. Vinh, Han Ting Wang, Vincent Mooser.

**Methodology:** Karine Tremblay, Simon Rousseau, Ma'n H. Zawati, Daniel Auld, Michaël Chassé, Emilia Liana Falcone, Nicolas Gauthier, Yann Joly, Daniel E. Kaufmann, Catherine Laprise, François Maltais, Anne-Marie Mes-Masson, Alexandre Montpetit, Alain Piché, J. Brent Richards, Sze Man Tse, Alexis F. Turgeon, Donald C. Vinh, Vincent Mooser.

**Project administration:** Karine Tremblay, Simon Rousseau, Ma'n H. Zawati, Daniel Auld, Michaël Chassé, Daniel Coderre, Emilia Liana Falcone, Nicolas Gauthier, Nathalie Grand-vaux, François Gros-Louis, Carole Jabet, Yann Joly, Daniel E. Kaufmann, Catherine Laprise, Catherine Larochelle, François Maltais, Anne-Marie Mes-Masson, Alexandre Montpetit, Alain Piché, J. Brent Richards, Sze Man Tse, Alexis F. Turgeon, Gustavo Turecki, Donald C. Vinh, Han Ting Wang, Vincent Mooser.

**Resources:** Karine Tremblay, Simon Rousseau, Ma'n H. Zawati, Michaël Chassé, Daniel Coderre, Carole Jabet, Daniel E. Kaufmann, Anne-Marie Mes-Masson, Alain Piché, J. Brent Richards, Sze Man Tse, Alexis F. Turgeon, Gustavo Turecki, Vincent Mooser.

**Software:** Michaël Chassé.

**Supervision:** Karine Tremblay, Simon Rousseau, Ma'n H. Zawati, Daniel Auld, Michaël Chassé, Emilia Liana Falcone, Nicolas Gauthier, Daniel E. Kaufmann, François Maltais, Alain Piché, J. Brent Richards, Sze Man Tse, Alexis F. Turgeon, Gustavo Turecki, Han Ting Wang, Vincent Mooser.

**Validation:** Karine Tremblay, Simon Rousseau, Ma'n H. Zawati, Michaël Chassé, Catherine Larochelle, J. Brent Richards.

**Visualization:** Karine Tremblay, Michaël Chassé, Catherine Larochelle, J. Brent Richards, Vincent Mooser.

**Writing – original draft:** Karine Tremblay, Simon Rousseau, Ma'n H. Zawati, Sze Man Tse.

**Writing – review & editing:** Karine Tremblay, Simon Rousseau, Ma'n H. Zawati, Daniel Auld, Michaël Chassé, Emilia Liana Falcone, Nicolas Gauthier, Nathalie Grandvaux, François Gros-Louis, Yann Joly, Daniel E. Kaufmann, Catherine Laprise, Catherine Larochelle, François Maltais, Alexandre Montpetit, Alain Piché, J. Brent Richards, Sze Man Tse, Donald C. Vinh, Vincent Mooser.

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
