## [Decision Letter · Decision Letter 0]

5 Mar 2021

PONE-D-20-40382

The Biobanque quebecoise de la COVID-19 (BQC19)- A case-control bioresource to prospectively study the clinical and biological determinants of COVID-19 clinical trajectories

PLOS ONE

Dear Dr. Rousseau,

Thank you for submitting your manuscript to PLOS ONE. After careful consideration, we feel that it has merit but does not fully meet PLOS ONE’s publication criteria as it currently stands. Therefore, we invite you to submit a revised version of the manuscript that addresses the points raised during the review process.

We look forward to receiving your revised manuscript.

Kind regards,

John S Lambert

Academic Editor

PLOS ONE

Journal Requirements:

3. One of the noted authors is a group or consortium (BQC19). In addition to naming the author group, please list the individual authors and affiliations within this group in the acknowledgments section of your manuscript. Please also indicate clearly a lead author for this group along with a contact email address.

Reviewers' comments:

Reviewer's Responses to Questions

**Comments to the Author**

1. Is the manuscript technically sound, and do the data support the conclusions?

Reviewer #1: Yes

2. Has the statistical analysis been performed appropriately and rigorously? 

Reviewer #1: N/A

3. Have the authors made all data underlying the findings in their manuscript fully available?

Reviewer #1: Yes

4. Is the manuscript presented in an intelligible fashion and written in standard English?

Reviewer #1: Yes

5. Review Comments to the Author

Reviewer #1: The manuscript "The Biobanque québécoise de la COVID-19 (BQC19) – A case-control bioresource to

prospectively study the clinical and biological determinants of COVID-19 clinical trajectories" by Rousseau and colleagues describes the excellent work that has been completed by the authors and collaborators to assemble a biobank of samples and data for use in Covid-19 Research.

Their efforts have created a resource which will doubtless drive significant research output over the coming years. They should be congratulated on their efforts and should disseminate news of their achievement widely to ensure patients, the community, wider Quebec society and the international research community see the resource they have assembled.

Notwithstanding this, there are a number of points in the manuiscript that the authors shoudl address.

1. Methodology

Throughout the manuscriot the authors use the terms "case-control" and "cohort" interchangeably. Indeed following my review i am still unsure if this is a case control studty as the name suggest. It reads like a cohort study (which includes negatives controls). The importance of this point is not solely academic. It is vital as it underpins the ambition of the programme and the type of studies it will facilitate

2. The authors use the term "high quality" to describe the samples repeatedly. Howver they do not explain what they mean by this. What are the standards, what assessments are done, what is the quality system.

3. A number of standards for biobanks have emerged including ISO, and it would be helfpul for the authors to describe how their facilities / biobank complies to these standards

4. Significantly more information should be provided on Pre-analytical quality control. What tests will be done, what pre-analytical variables are collected. Are SPREC codes used ? this will help answer the questions above regarding quality.

5. The authors do not include Respiratory samples in the biobank, thus reducing the opportunity for viral genome sequencing as welll as respiratory pathogen host interaction studies. Perhaps the authors could comment on this

6. On page 10 the authors point out that "the consent process is specific to its own institution". How can the biobank be sure of standardisation?

7. The consent discussion on page 10, which is vital, moves into a dicussion about SOPs- i think these sections should be seperated.

8 On page 12 the authors say, "Usage of BQC19 samples and data is only possible if aligned with a participants consent". the authors shoulkd describe this in more detail including the consent types, how these are tracked, how is it checked to make sure the patients wishes are complied with... what is the process?

9. the results and discussion section is not well assembled . The main points that need to be presented here are

a) What % (and N) of the total number of hopitalised patients, were invited to participate

b) What % (and N) of those who were invited to participate gave consent

c) What % (and N) of those who consented have full samples sets ( even for acute phase)

d) There is no demographic data presented- this would be helpful to gauge how successful the biobank has been in collecting a representative sample of Quebecoise with covid-19

e) It would be helpful to understand the perfomance of the different sites... where did the samples come from, the relative perfomances of different sites. This is vital to assessing the performance of the network

10. he authors shoudl describe their audit and compliance plans, to ensure sites comply with the BQC19 SOPS- in the absence of this oversight, the concern is that this ill remain as just a sample collection from multiple sites, as opposed to an integrated biobank

11 It would be helpoful if the authors included within the discussion, theiur thoughts on the demand for data so far and what they anticipate will be the demand for data and samples going forward.

6. PLOS authors have the option to publish the peer review history of their article (what does this mean?). If published, this will include your full peer review and any attached files.

Reviewer #1: No

---

## [Author Response · Author response to Decision Letter 0]

30 Mar 2021

Comments to the Author

Reviewer #1 

General comment:

The manuscript "The Biobanque québécoise de la COVID-19 (BQC19) – A case-control bioresource to prospectively study the clinical and biological determinants of COVID-19 clinical trajectories" by Rousseau and colleagues describes the excellent work that has been completed by the authors and collaborators to assemble a biobank of samples and data for use in Covid-19 Research.

Their efforts have created a resource which will doubtless drive significant research output over the coming years. They should be congratulated on their efforts and should disseminate news of their achievement widely to ensure patients, the community, wider Quebec society and the international research community see the resource they have assembled.

Notwithstanding this, there are a number of points in the manuscript that the authors should address.

Response:

We are thankful for these encouragements and are working hard at promoting BQC19 as a resource to further research into COVID-19. Central to our dissemination strategy is the present research paper that we hope will reach a wide audience encouraging them to access our biological material, clinical and experimental data available for researchers. This is why we are grateful for the comments below that help improving our manuscript. In addition, we have updated the information with current enrollment data and the latest information on our procedures as part of the review process. 

Comment 1

Methodology - Throughout the manuscript the authors use the terms "case-control" and "cohort" interchangeably. Indeed following my review i am still unsure if this is a case control study as the name suggest. It reads like a cohort study (which includes negatives controls). The importance of this point is not solely academic. It is vital as it underpins the ambition of the programme and the type of studies it will facilitate.

Response: 

We agree and have replaced the term case-control bioresources with cohort (that includes negative controls) as it better reflects the nature of BQC19’s project.

Comment 2 

The authors use the term "high quality" to describe the samples repeatedly. However they do not explain what they mean by this. What are the standards, what assessments are done, what is the quality system.

Response: 

The reviewer is correct in pointing out that we did not define sufficiently what was meant by high quality biosamples. We have now included a new subsection (Biosamples pre-analytical quality control, page 12, lines 247-259) and table (Table 3, p.25) in the result section that state clearly the measures put in place to ensure the quality of the collected biosamples in addition to the standardization of protocols. The new paragraph is re-transcribed here:

Biosamples pre-analytical quality control

In order to ensure consistency in the preparation of biosamples collected for BQC19, all participating sites use the same SOPs (available at BQC19.ca), clearly detailing the exact protocol to be followed, including the type of primary container to be used (Table 3). In terms of pre-analytical quality assessment, we document the following information for each sample collected: date of collection, location of sample, associated barcode(s), SOPs used, time elapsed to biobank, precisions/explanations on delays, name of sample’s handler. Our protocol details that samples should ideally be processed in less than 6h from collection, the actual time (pre-centrifugation time) is recorded for each sample in the biobank management software. In addition, for PBMCs we document cell concentration, number of cells in sample (total cells count), freezing medium; for nucleic acids isolations, we document extraction date, extracted from [biological specimen], extraction method, diluent, absorbance measurements at 260nm, 280nm and 320nm, 260/280 ratio, dilution factor, concentration, total amount. 

Comment 3 

A number of standards for biobanks have emerged including ISO, and it would be helfpul for the authors to describe how their facilities / biobank complies to these standards.

Response: 

A key feature of BQC19 was its development during a sanitary emergency period, with the elaboration of the ethical and legal framework and study protocols done as quickly and efficiently as possible to enable the collection of biological material and clinical data of the first occurring wave of infection (March 2020 in Québec). A year later, in view of the evolution of the pandemic, the appearances of new variants, the recruitment of vaccinated participants, the emphasis is still on being as responsive as possible to the continuously evolving pandemic situation. As the biobank is still in its development phase, it is too early to seek out certification for international standards. Following the reviewer’s comment, we asked the BQC19 coordinator to look into the ISBER biobank self-assessment tool as a path towards certification. At this early stage, it was not possible to complete enough of the 152 questions to obtain sufficient information. However, we do plan to seek out certification by having each site fill in the questionnaire and return it to us in the near future. 

Having said that, the elaboration of the BQC19 has addressed from the get-go some of the standards described below:

1-Background information: The management framework, which is freely and publicly available on BQC19’s website, provide detailed information about the mission, structure and procedures of BQC19. 

2-Repository planning considerations, facilities, storage equipment/environment: As appendices to the management framework, each site have given a detailed description of where samples and data are stored. This information has been reviewed by a coordinating Institutional Ethics Review Board (CHUM). We are holding discussion with provincial biobanking facilities for option on long-term storage of the thousand’s samples generated by the project.

3-Quality management: Quality management is first established by providing detailed SOPs that are not only distributed to each site but also publicly available via BQC19’s website. The biobank coordinator is responsible for ensuring that all updates to SOPs are distributed and put in place via the network of each site coordinators. Moreover, the biobank coordinator monitored the quality of the clinical data entry provided by each site. Our biobank management software captures numerous fields related to pre-analytical quality control as detailed in response to Comment 2 above.

4-Safety & training: The required training and safety procedures are detailed in BQC19’s management framework. The biobank coordinator ensures that a copy of supporting documents is sent to BQC19 from each participating institution.

5-Record management: The multicentric design of BQC19 is managed using the Nagano platform that enables the efficient distribution of updated documents pertaining to each participating sites. Moreover, the biobank has also a full-time manager, responsible for records keeping of the steering’s committee minutes, the budget (current and forecast), monthly reports to the governing board and yearly report to funding organizations. The biobank coordinator, also appointed full-time, ensures that BQC19 receives copies of relevant records from each site, pertaining to procedures, training and access. The BQC19 access officer is in charge of the web platform that manages access requests and record-keeping of the documentation pertaining to access, including minutes from the independent access committee.

6-Cost management: The overall budget is established by the steering committee and then approved by the governance committee. The biobank manager and coordinator work in tandem to establish detailed budgets, ensure each site adheres to guidelines and that expenses are tracked.

7-Legal and ethical issues for biospecimens: A member of the steering committee, Ma’n Zawati is responsible for legal and ethics issues, not only about biosamples but more generally as well. He works in tandem with the Institutional Ethics Review Board to ensure that we follow an appropriate legal and ethical framework.

8-Specimen access, utilization and destruction: Access procedures are developed based on whether biological material is required or not. Access procedures are established by BQC19’s steering committee, but it is an independent access committee that provides the evaluations of the request. The access officer is responsible for coordinating the access process. We have detailed Data and Material Transfer Agreements (D/MTAs) that are sent to end-users that define clearly legal and ethical obligation pertaining to their request. The process for samples destruction is detailed in BQC19’s management framework. 

Comment 4 

Significantly more information should be provided on Pre-analytical quality control. What tests will be done, what pre-analytical variables are collected. Are SPREC codes used ? this will help answer the questions above regarding quality.

Response: 

Although we have not used the SPREC coding scheme per se (but we are adapting our biobank management software to include them), we have the information relating to each of the sample quality variable captured in our database. By looking at Table 3, the only information captured by SPREC codes that we do not have in BQC19 is the post-centrifugation time to storage. The other variables are there but coded differently. As stated in our response to Comment 2 above, a new paragraph and table has been provided in the manuscript in order to enable readers to assess pre-analytical quality control information much more readily. Moreover, a particularity of BQC19’s is the generation of analytical data from biosamples made available to all researchers via the access process. The analytical data is not only itself critically evaluated for quality internally but also informs us on the quality of the biosamples in the collection. 

Comment 5 

The authors do not include Respiratory samples in the biobank, thus reducing the opportunity for viral genome sequencing as welll as respiratory pathogen host interaction studies. Perhaps the authors could comment on this.

Response: 

In the initial design we had envisaged collecting minimally nasal swabs and, in institutions where it was feasible, airway secretions. However, the particular context of the pandemic, especially during the first wave of infection, made this either difficult or simply not feasible. For example, no nasal swabs were available for research at that time, as they were all required for testing. Moreover, precautionary measures needed to be taken for the collection of airway-derived biosamples meant that only few centres were willing/equipped to handle the procedures. This was compounded by the limited supplies of personnel protection equipment (PPE). 

That said, we agree with the reviewer that there can be tremendous value in joining information from viral sequencing to the BQC19 participants. To directly address this point, since last summer we entered into discussion with the provincial facilities that performs analyses (sequencing) on all nasal swabs collected by health authorities (LSPQ), to establish a collaboration in order to link viral sequences to BQC19’s participants. This key piece of information can be found in the manuscript in Future directions subsection, networking (p.19-20, lines 416-422). 

Comment 6 

On page 10 the authors point out that "the consent process is specific to its own institution". How can the biobank be sure of standardisation?

Response:

Thank you for this comment. This specific sentence was removed from the paragraph as it is confusing. All BQC19 sites follow common SOPs (and abide by the same principles) to ensure standardization, something that is mentioned later in the same paragraph.

Comment 7 

The consent discussion on page 10, which is vital, moves into a discussion about SOPs- i think these sections should be separated.

Response:

As suggested by the reviewer, the Consent discussion has been broken into two sections as suggested, one on considerations and the other on procedures.

Comment 8 

On page 12 the authors say, "Usage of BQC19 samples and data is only possible if aligned with a participants consent". the authors should describe this in more detail including the consent types, how these are tracked, how is it checked to make sure the patients wishes are complied with... what is the process?

Response: 

Thank you for this comment. Indeed, the sentence as it is does not provide a full picture. A sentence was added to explain that this is actually made possible by ensuring that the access process as well as the terms and conditions of any future use of data and samples respect the general permissions consented to by participants (p.12, lines 262-263). Given that the consent process is common across all sites, the access principles are consistent as well.

Comment 9 

The results and discussion section is not well assembled. The main points that need to be presented here are

a) What % (and N) of the total number of hopitalised patients, were invited to participate

b) What % (and N) of those who were invited to participate gave consent

c) What % (and N) of those who consented have full samples sets (even for acute phase)

d) There is no demographic data presented- this would be helpful to gauge how successful the biobank has been in collecting a representative sample of Quebecoise with covid-19

e) It would be helpful to understand the performance of the different sites... where did the samples come from, the relative performances of different sites. This is vital to assessing the performance of the network.

Response:

As requested by the reviewer, we have collected and added these important data to the Results & Discussion section of the manuscript. More precisely:

Requests a) and b): In order to well report BQC19 participation acceptance rates, we added a sub-section named “Enrollment statistics” (p.14, lines 286-302). The new paragraph is re-transcribed here:

Enrollment statistics

For the hospitalized cohort, we report a 75.4% acceptance rate (2,1271 out of 2,878 invited to participate excluding patients who were discharged or scheduled to be discharged, deceased, incapacited, or inadmitted to care units without planned blood sampling; total form eight sites). In the non-hospitalized cohort, we report an acceptance rate of 80.2% (616 out of 768 invited to participate; total from five sites). The higher success rate in the non-hospitalized cohort may be explained by different enrolling strategies across sites (e.g. two of the sites used public advertising which includes a voluntarism bias). In term of dropout rates, we report 3.4% in the hospitalized cohort (73 dropouts out of 2171 enrolled participants) and a 0.5% rate in the non-hospitalized cohort (3 dropouts out of 616 enrolled participants). Finally, for both cohorts, reported reasons for refusal or study dropout include: “no interest”, “no benefit”, “don’t believe in research purposes”, “no time for follow-up”, “surrogate refusal”, “health related reasons/age”, “SARS-CoV-2 negative patient who though their participation wasn’t important”, “fear about the future uses of their data (or their children’s data)”, “parents’ fear of harming their children”, “unwillingness to move or to give more blood for follow-up visits”, “communication/understanding issues”, “difficulty in taking blood samples”, and “overburdened by hospitalization and their clinical follow-up/worried enough”. 

Request c): A precision about the available full sample sets has been added in the text manuscript (p.15, lines 314-318) as following: 

For the hospitalized cohort, where participants have been enrolled at the time of hospital admission (Day 0), the full sample sets currently available at each timepoint are shown in Figure 6A. For the non-hospitalized cohort, the full sample sets are available for all participants at Day 180 post-infection (Figure 6B). However, in this cohort, some participants may have been enrolled at Day 30 or Day 90 post-infection.

Request d): The biobank was not designed to capture an accurate reflection of the demographic representation of the province’s population, but to recruit cases as quickly as possible, to support research during the sanitary emergency. Available information on BQC19 demographics and remark about the representativity of the enrolled participants have been added in a separate paragraph of the BQC19 participants' characteristics and available data sub-section (p.15, lines 321-327). The new text is re-transcribed here:

The participants currently included in BQC19’s database are distributed in four Québec Health Regions: 1808 (78.6%) from Montréal (five enrolling sites); 291 (12.7%) from Estrie (one enrolling site); 144 (6.3%) from the Saguenay-Lac-Saint-Jean (one enrolling site); and 57 (2.5%) from the Capitale-Nationale (two enrolling sites). This is not an accurate reflection of the demographic representation of the province’s population, which was not the goal of this biobank, but rather to recruit participants as quickly as possible, to support research during the sanitary emergency. 

Request e): Since all sites did not begin recruiting at the same time, that all sites are in active enrollment and that not all sites are located epidemiologic “hot spots” of Québec province wave infection, comparing their performance based on their enrollment statistics does not provide an accurate picture of the substantial commitment made each site. Therefore, we prefer reporting the aggregate data. However, data added in regard to Request d) above give the reader more precision on the Québec distribution of the BQC19 participants, which indirectly provide an indicator of sites performance. 

Comment 10 

The authors should describe their audit and compliance plans, to ensure sites comply with the BQC19 SOPS- in the absence of this oversight, the concern is that this ill remain as just a sample collection from multiple sites, as opposed to an integrated biobank

Response:

As mentioned in response to Comment 3 above, the biobank coordinator ensures compliance of each site. First, she makes sure each site has the correct documentation. Audit is made in two ways: 1) the first one is via the biobank management software that captures key information about respect of procedures. Moreover, the last portion of fees paid for each participant recruited to the study are only to each site once the biobank coordinator has reviewed the data, found it complete and in accordance with procedures. This is done at the end of every other month. Moreover, each site must provide documentation supporting safety and training of personnel on good clinical and laboratory practices, facilities assigned to storage of material and data related to BQC19’s project and a yearly report of activities.

Comment 11 

It would be helpful if the authors included within the discussion, their thoughts on the demand for data so far and what they anticipate will be the demand for data and samples going forward.

Response:

We currently have received 11 requests for access to data only, 10 of them having been accepted. The process for access to data is continuous. Moreover, we have received 3 requests for biological material access, with 2 accepted during the first organized call for access (there are at least 3 fixed dates for submission to access biological material per calendar year currently). We have amended our conclusion to include the following in the manuscript text (p.20, lines 426-430): “We have already provided access to data or biological material to more than a dozen investigators in the first few months of operations. By providing access to the research community to clinical data as well as data derived from in-depth multi-omic analyses on the first 2000 samples, we are forecasting (and encouraging) an exponential increase in requests of this valuable and non-depletable resource.”

---

## [Editor Report · Decision Letter 1]

5 Apr 2021

The Biobanque québécoise de la COVID-19 (BQC19) – A cohort to prospectively study the clinical and biological determinants of COVID-19 clinical trajectories

PONE-D-20-40382R1

Dear Dr. Rousseau,

We’re pleased to inform you that your manuscript has been judged scientifically suitable for publication and will be formally accepted for publication once it meets all outstanding technical requirements.

Kind regards,

John S Lambert

Academic Editor

PLOS ONE

Additional Editor Comments (optional):

responses have been provided and now adequate for publication
---

## [Editor Report · Acceptance letter]

7 May 2021

PONE-D-20-40382R1 

The Biobanque québécoise de la COVID-19 (BQC19) – A cohort to prospectively study the clinical and biological determinants of COVID-19 clinical trajectorie 

Dear Dr. Rousseau:

I'm pleased to inform you that your manuscript has been deemed suitable for publication in PLOS ONE. Congratulations! Your manuscript is now with our production department. 

Kind regards, 

on behalf of

Dr. John S Lambert 

Academic Editor

PLOS ONE